# Sex-Specific Alterations in Dopamine Metabolism in the Brain after Methamphetamine Self-Administration

**DOI:** 10.3390/ijms23084353

**Published:** 2022-04-14

**Authors:** Atul P. Daiwile, Patricia Sullivan, Subramaniam Jayanthi, David S. Goldstein, Jean Lud Cadet

**Affiliations:** 1Molecular Neuropsychiatry Research Branch, NIDA Intramural Research Program, National Institutes of Health (NIH), Baltimore, MD 21224, USA; atul.daiwile@nih.gov (A.P.D.); sjayanth@intra.nida.nih.gov (S.J.); 2Autonomic Medicine Section, NINDS Intramural Research Program, National Institutes of Health (NIH), Bethesda, MD 20892, USA; psullivan1@ninds.nih.gov (P.S.); goldsteind@ninds.nih.gov (D.S.G.)

**Keywords:** methamphetamine, sex differences, dopamine, DA metabolites, mesocorticolimbic

## Abstract

Methamphetamine (METH) use disorder affects both sexes, with sex differences occurring in behavioral, structural, and biochemical consequences. The molecular mechanisms underlying these differences are unclear. Herein, we used a rat model to identify potential sex differences in the effects of METH on brain dopaminergic systems. Rats were trained to self-administer METH for 20 days, and a cue-induced drug-seeking test was performed on withdrawal days 3 and 30. Dopamine and its metabolites were measured in the prefrontal cortex (PFC), nucleus accumbens (NAc), dorsal striatum (dSTR), and hippocampus (HIP). Irrespective of conditions, in comparison to females, male rats showed increased 3,4-dihydroxyphenylalanine (DOPA) in the PFC, dSTR, and HIP; increased cys-dopamine in NAc; and increased 3,4-dihydroxyphenylethanol (DOPET) and 3,4-dihydroxyphenylacetic acid (DOPAC) in dSTR. Males also showed METH-associated decreases in DA levels in the HIP but increases in the NAc. Female rats showed METH-associated decreases in DA, DOPAL, and DOPAC levels in the PFC but increases in DOPET and DOPAC levels in the HIP. Both sexes showed METH-associated decreases in NAc DA metabolites. Together, these data document sex differences in METH SA-induced changes in DA metabolism. These observations provide further support for using sex as an essential variable when discussing therapeutic approaches against METH use disorder in humans.

## 1. Introduction

Methamphetamine (METH) is the second-most-used psychoactive substance. Its consumption is associated with severe negative consequences on health and socioeconomic status in both male and female users [1,2,3,4,5,6,7,8,9,10,11,12,13,14,15]. Male users usually take more METH than females [9,10,16,17], whereas women present with greater dependency on the drug and relapse more frequently during treatment. Women appear to also suffer from a higher prevalence of psychiatric complications [6,18,19,20,21,22].

Although sex differences in behavioral endpoints associated with METH use have also been documented in some preclinical studies [23,24,25,26,27,28], there have been few studies that have reported on potential biochemical or molecular consequences that might accompany METH self-administration (SA). For example, male rats with a history of METH SA exhibited increased hippocampal BDNF protein levels [29], higher co-localization of 5-hydroxytryptamine 2C receptor (5-HT2CR) with parvalbumin interneurons in the orbitofrontal cortex [30], but decreased serotonin transporter (5HTT) levels in the infralimbic cortex (IL) [31] in comparison to females. On the other hand, female METH SA rats showed higher c-Fos immunoreactivity in the cortex, dorsal striatum (dSTR), nucleus accumbens (NAc), and amygdala in comparison to males [32]. More recently, Daiwile et al. (2019, 2021) have documented substantial sex differences in METH SA-associated changes in gene expression in various brain regions [27,33].

An important theoretical framework for substance use disorders (SUDs) revolves around brain dopaminergic systems that play substantial roles in reward, learning and memory, and decision making [34,35,36,37,38,39]. Distinct but interconnected brain regions that contain dopamine (DA) projections from the midbrain cell bodies located in the ventral tegmental area (VTA) and substantia nigra pars compacta (SNpc), and other projections to subcortical and cortical brain regions appear to be involved in various aspects of SUD diatheses [40,41,42,43]. As part of our efforts to identify biochemical consequences of exposure to METH SA, the present study was conducted to measure the levels of DA and its metabolites in the prefrontal cortex (PFC), nucleus accumbens (NAc), dorsal striatum (dSTR), and hippocampus (HIP) of female and male rats after a month of withdrawal from 20 days of exposure to METH SA.

## 2. Results

### 2.1. Sex Differences in METH Intake

We analyzed daily METH intake patterns between males and females after 20 days of self-administration. As previously published [27,33], two-way repeated measure ANOVA found significant effects of sex (F (1, 30) = 7.214, *p* = 0.0117), METH intake (F (19, 570) = 18.23, *p* < 0.0001), and sex x METH intake interaction (F (19, 570) = 3.562, *p* < 0.0001). Post-hoc analysis showed that males self-administered more METH when compared with females (Figure 1A). Both female and male METH rats increased their METH intake over 20 days when compared to their first day of METH intake. We also compared female saline and male saline, which showed significant effects of saline intake (F (19, 190) = 2.249, *p* = 0.0031), but no effect of sex (F (1, 10) = 2.804, *p* = 0.1250) and their interaction (F (19, 190) = 1.347, *p* = 0.1583). Both male and female rats decreased their saline intake over time (see Appendix A for graphic illustration).

In addition, the analysis showed that individual female and male rats did not self-administer METH to similar degrees because some animals escalated their METH intake compared to their first day of METH intake, whereas others took similar amounts of METH from the beginning to the end of the behavioral experiment [27,33]. Rats that escalated their METH intake were labeled ‘high METH takers’, whereas those that did not exhibit escalation were called ‘low METH takers’ (Figure 1B). When we compared female and male high METH takers statistically, we found no effects of sex (F (1, 17) = 2.278, *p* = 0.1496), but significant effects of METH intake (F (19, 323) = 17.85, *p* < 0.0001) and their interaction (F (19, 323) = 3.344, *p* < 0.0001). Two-way ANOVA for female and male low METH taker also demonstrated significant effects of METH intake (F (19, 209) = 2.723, *p* = 0.0003), but no effect of sex (F (1, 11) = 2.462, *p* = 0.1449) and their interaction (F (19, 209) = 0.9923, *p* = 0.4715). Comparison between male high and low METH-taker rats showed significant effects of group (F (1, 13) = 11.74, *p* = 0.0045), METH intake (F (19, 247) = 7.408, *p* < 0.0001), but not their interaction (F (19, 247) = 1.363, *p* = 0.1456). Post-hoc analysis revealed that there were significant differences in METH intake between high and low METH takers from day 5 onward. Male high METH takers also showed significant increases in their METH intake from day 2 onward compared to day 1; similar effects were not observed in low METH takers. Analysis between female high and low meth takers revealed significant effects of group (F (1, 15) = 17.61, *p* = 0.0008), METH intake (F (19, 285) = 10.03, *p* < 0.0001), and their interaction (F (19, 285) = 3.340, *p* < 0.0001). Post-hoc analysis revealed that female high METH takers self-administered more METH from day 5 onward compared to low METH takers. Female high METH takers, but not low takers, increased their METH intake from day 4 onward compared to day 1.

As reported earlier [27,33], there were no sex differences in METH-seeking behavior after withdrawal from METH SA, with both female and male METH rats increased their active lever responding on WD30 compared to WD3 (Figure 1C). Three-way ANOVA revealed significant effects of withdrawal day (F (1, 39) = 11.09, *p* = 0.0019) and group (saline vs. METH) (F (1, 39) = 16.70, *p* = 0.0002), but there were no effects of sex (F (1, 39) = 0.7130, *p* = 0.4036) or overall interaction (withdrawal day x sex x saline vs. METH) (F (1, 39) = 0.3498, *p* = 0.5577).

### 2.2. Levels of Dopamine and Its Metabolites

Previous publications from the Cadet’s laboratory have reported that contingent and non-contingent administration of METH can impact DA levels in the brain of male rodents [44,45,46,47,48]. However, there have been only a few studies that have examined the effects of sex in METH-induced changes in striatal DA and its metabolites in mice [49,50,51]. These studies have reported that METH impacts striatal DA levels in a sexually dimorphic fashion [49,50,51]. We thus wanted to know if METH SA might have different consequences on DA and its metabolites in a brain region- and sex-dependent manner.

#### 2.2.1. Prefrontal Cortex (PFC)

The PFC is known to participate in the regulation of self-control, planning, decision making, awareness, and moderating social behavior, all of which are thought to contribute to the clinical course of substance use disorders (SUDs) [40,52]. We thus measured and compared the levels of DOPA (precursor) as well as DA and its metabolites in drug-naïve and METH-exposed female and male rats. ANOVA revealed significant effects of sex (F (1, 18) = 25.04, *p* < 0.0001), METH intake (F (2, 19) = 7.596, *p* = 0.0038), and their interaction (F (2, 19) = 4.430, *p* = 0.0264) on DOPA levels in the PFC. Post-hoc analysis showed that male rats had higher levels of PFC DOPA than female rats, irrespective of conditions. Interestingly, only male METH low takers showed significant increases in DOPA levels when compared to male controls and METH high takers (Figure 2A). The quinone derivative of DOPA, cys_DOPA, which is produced via DOPA oxidation exhibited no effects of sex (F (1, 36) = 0.1837, *p* = 0.6708), METH intake (F (2, 36) = 1.878, *p* = 0.1675), or their interaction (F (2, 36) = 1.083, *p* = 0.3492) (Figure 2B).

Two-way ANOVA for DA levels revealed significant effects of sex (F (1, 33) = 25.70, *p* < 0.0001), METH intake (F (2, 33) = 13.50, *p* < 0.0001), and their interaction (F (2, 33) = 17.92, *p* < 0.00010). In addition, female saline rats showed higher levels of DA when compared with males. Moreover, only female METH rats showed significant decreases in the levels of DA compared to control animals (Figure 2C). Although this suggestion will need to be tested, the decrease in DA levels in female rats might be related to increased accumulation of METH in the PFC of female rats. These observations might also be related to toxicity related to pre-existing high DA levels in the female PFC. This suggestion is consistent in part with the results of Becker et al. (1982), who had reported higher levels of amphetamine in the whole brains of female rats after systemic administration [53].

Statistical analysis of the effects of METH on DOPAL, an oxidative DA metabolite, revealed significant effects of METH intake (F (2, 33) = 5.660, *p* = 0.0077) and sex x METH intake interaction (F (2, 33) = 4.646, *p* = 0.0167) but no effect of sex (F (1, 33) = 1.161, 0.2891). Female saline rats also displayed higher levels of DOPAL than males. In a fashion similar to DA levels, only female METH SA rats showed decreased levels of DOPAL compared to controls (Figure 2D).

Analysis of the effects of METH on PFC DOPET revealed significant effect of METH intake (F (2, 35) = 23.84, *p* < 0.0001) but no effect of sex (F (1, 35) = 0.8991, *p* = 0.3495) or their interaction (F (2, 35) = 2.848, *p* = 0.0715). Post-hoc analysis shows that female high METH takers had higher levels of DOPET when compared with low METH takers and control rats. Both low and high male METH takers showed higher DOPET levels in comparison to controls. High METH-taker rats also had higher DOPET levels compared to low METH-taker rats (Figure 2E).

Statistical analysis for the effects of METH on DOPAC levels showed significant effects of METH intake x sex interaction (F (2, 34) = 5.443, *p* = 0.0089). Similar to the situation for DA levels, female saline rats showed higher DOPAC levels compared to males. In addition, only female METH rats showed significant decreases in DOPAC levels after METH in comparison to control levels (Figure 2F), supporting the notion mentioned above that METH might have accumulated to a higher concentration in the PFC of female rats to cause toxicity.

#### 2.2.2. Nucleus Accumbens (NAc)

The NAc is an important nodal point in the brain reward circuitry that participates in the development and maintenance of drug-taking behaviors [35,54,55,56,57]. In the NAc, we found that male control rats exhibited higher DOPA levels than females. ANOVA identified significant effects of sex (F (1, 36) = 21.39, *p* < 0.0001), METH intake (F (2, 36) = 71.93, *p* < 0.0001), and their interaction (F (2, 36) = 7.642, *p* = 0.0017). Post-hoc analysis showed that both female and male METH low and high taker rats had a reduction in DOPA levels after METH SA in comparison to respective controls (Figure 3A). In the case of cys_DOPA, female control rats displayed higher levels than male rats. There was also a significant METH intake vs. sex interaction (F (2, 18) = 7.742, *p* = 0.0037). This was due to the fact that female high METH-taker rats experienced a reduction in cys-DOPA levels, whereas male high METH takers had increased cys-DOPA levels in comparison to control and low METH takers (Figure 3B).

DA levels were increased in male high METH takers but not in females in comparison to controls (Figure 3C). ANOVA for DA revealed a significant effect of METH intake vs. sex interaction (F (2, 18) = 3.506, *p* = 0.05). ANOVA for the effects of METH on the quinone DA derivative, cys_DA, showed significant effects of METH intake (F (2, 18) = 51.25, *p* < 0.0001), but no effect of sex (F (1, 18) = 2.702, *p* = 0.1176) and their interaction (F (2, 18) = 1.515, *p* = 0.2464). Moreover, cys-DA levels were decreased in female and male low and high METH takers in comparison to their respective controls, with female high METH takers experiencing greater decreases in cys_DA levels than female low METH takers (Figure 3D).

DOPAL levels were decreased in female and male low and high METH takers compared to their controls. ANOVA found significant effect of METH intake (F (2, 18) = 34.79, *p* < 0.0001), but no effect of sex (F (1, 18) = 0.1515, *p* = 0.7016) and METH intake vs. sex interaction (F (2, 18) = 3.262, *p* = 0.0618) (Figure 3E).

In general, male rats exhibited higher levels of DOPET and DOPAC compared to female rats (Figure 3F,G). ANOVA for DOPET levels showed effects of sex (F (1, 18) = 10.96, *p* = 0.0039), METH intake (F (2, 18) = 14.39, *p* = 0.0002), but not their interaction (F (2, 18) = 2.844, *p* = 0.0844). Post-hoc analysis revealed lower DOPET levels in female and male METH rats compared to their controls (Figure 3F). For DOPAC, we found significant effects of sex (F (1, 18) = 10.52, *p* = 0.0045), METH intake (F (2, 17) = 7.887, *p* = 0.0038), and their interaction (F (2, 17) = 3.858, *p* = 0.0415). Moreover, low and high male METH takers exhibited decreased DOPAC levels compared to control animals, with similar decreases being seen only in female high METH takers (Figure 3G).

#### 2.2.3. Dorsal Striatum (dSTR)

The dorsal striatum participates in the behavioral consequences that result from manipulations of the reward circuit [38,58]. It was therefore important to know if chronic METH SA might perturb the levels of DA and its metabolites in that structure. Starting with DOPA levels, we found significant effects of sex (F (1, 17) = 48.29, *p* < 0.0001) and sex x METH intake interaction (F (2, 20) = 4.688, *p* = 0.0214), but no effect of METH intake (F (2, 20) = 2.249, *p* = 0.1316). Saline and male METH self-administering rats had higher DOPA levels than respective female groups. In addition, only male METH rats showed significant decreases in DOPA levels in comparison to male controls (Figure 4A). Moreover, there were no significant changes in cys_DOPA levels after METH in female and male rats: METH intake (F (2, 38) = 2.422, *p* = 0.1023), sex (F (1, 38) = 0.4235, *p* = 0.5191), or their interaction (F (2, 38) = 1.234, *p* = 0.3026) (Figure 4B).

DA levels remained unchanged after METH SA in female and male rats: sex (F (1, 18) = 2.445, *p* = 0.1353), METH intake (F (2, 20) = 0.1203, *p* = 0.8873), or their interaction (F (2, 20) = 2.471, *p* = 0.1099) (Figure 4C). The quinone DA derivative, cys_DA, showed significant effects of sex (F (1, 16) = 14.95, *p* = 0.0014), METH intake (F (2, 20) = 4.478, *p* = 0.0247), but not their interaction (F (2, 20) = 0.3541, *p* = 0.7061). Specifically, male METH rats exhibited higher cys_DA levels than female METH rats. In addition, male high METH takers showed decreased cys_DA levels in comparison to male low METH takers, but not from saline male rats (Figure 4D). Unexpectedly, there were no significant changes in the levels of the DA metabolites, DOPAL (Figure 4E), DOPET (Figure 4F), and DOPAC (Figure 4G) in the dorsal striatum of these rats. Together, all these observations suggest that the levels of METH might have reached similar levels in the dorsal striatum as reported previously for amphetamine by Becker et al. (1982) [53].

#### 2.2.4. Hippocampus (HIP)

The hippocampus (HIP) is important for learning and memory formation and also participates in the behavioral manifestations of SUDs [34,41,59]. We thus measured the effects of METH SA on DA and its metabolites in that structure. We found that all male rats expressed higher DOPA levels than female rats (Figure 5A). There were significant effects of sex (F (1, 18) = 51.50, *p* = <0.0001), METH intake (F (2, 20) = 7.940, *p* = 0.0029), and METH intake vs. sex interaction (F (2, 20) = 4.352, *p* = 0.0270). Male low METH takers showed decreased DOPA levels in comparison to controls, whereas male high METH takers showed increased DOPA levels in comparison to control and low METH takers (Figure 5A). The quinone derivative of DOPA, cys-DOPA, showed no effects of METH intake (F (2, 37) = 0.7032, *p* = 0.5015) or sex vs. METH intake interaction (F (2, 37) = 2.767, *p* = 0.0759). However, there was a sex effect (F (1, 37) = 6.512, *p* = 0.0150), with female low METH takers having higher levels of cys_DOPA compared to male counterparts (Figure 5B).

This is the first description that we are aware of that DA levels are higher in the hippocampus of male control rats compared to females (Figure 5C). There were significant effects of sex (F (1, 17) = 5.986, *p* = 0.0256) and sex vs. METH intake interaction (F (2, 20) = 3.375, *p* = 0.05) but not of METH intake (F (2, 20) = 1.129, *p* = 0.3431), with only male high METH takers showing decreased DA content when compared to control and male low METH takers (Figure 5C).

ANOVA for DOPAL revealed significant effects of METH intake vs. sex interaction (F (2, 17) = 6.863, *p* = 0.0065) in the HIP. Male control rats had higher DOPAL levels than females. Interestingly, female high METH takers had increased DOPAL levels, whereas male low and high male METH takers showed decreased DOPAL levels in comparison to their respective male controls. (Figure 5D).

Female low and high METH takers showed dose-dependent increases in DOPET levels compared to control. There were also significant effects of METH intake (F (2, 18) = 6.618, *p* = 0.0070) and sex vs. METH intake interaction (F (2, 18) = 4.877, *p* = 0.0203), but no effect of sex (F (1, 15) = 0.02875, *p* = 0.8676). This is due to the fact that DOPET levels in control female and male rats were comparable at baseline, but METH SA caused increased DOPET levels in female high METH takers compared to male high METH takers (Figure 5E).

Low and high female METH takers showed increased DOPAC levels in comparison to control rats, suggesting increased DA metabolism in these female rats. There were significant effects of METH intake (F (2, 20) = 3.794, *p* = 0.0401) and sex vs. METH interaction (F (2, 20) = 9.192, *p* = 0.0015), but not sex effect (F (1, 15) = 3.597, *p* = 0.0773). Similar to observations for HIP DA levels, male high METH takers showed a reduction in DOPAC levels compared to male low METH takers. Again, similar to observations with DA levels, male controls and low METH takers had higher DOPAC levels when compared to their female counterparts. After METH exposure, female high METH takers increased their DOPAC levels to those observed in male high takers (Figure 5F).

## 3. Discussion

The present study is the first to measure potential sex-related differences in DA and its metabolites in the projections of both nigrostriatal and mesocorticolimbic dopaminergic systems in rats that had either escalated or not escalated their intake of METH during a drug self-administration experiment. Our results showed that (i) male rats exhibited higher levels of DOPA in the PFC, dSTR, and HIP; of cys-DA in the dSTR; of DOPET and DOPAC in the NAc in comparison to female rats, irrespective of experimental conditions (saline or METH); (ii) female control rats displayed higher levels of DA, DOPAL, and DOPAC in the PFC and of cys_DOPA in the NAc in comparison to their male counterparts; (iii) male METH rats had decreased levels of DOPA in the PFC and dSTR, of cys_DA in the dSTR, and DA in the HIP; there were, however, increased levels of DOPA in the HIP and of DA in the NAc; (iv) female METH rats showed decreased levels of DA, DOPAL, and DOPAC in the PFC but increased levels of DOPET and DOPAC in the HIP; (v) both female and male METH rats showed elevated levels of DOPET in the PFC but decreased levels of DOPA, cys_DA, DOPAL, DOPET, and DOPAC in the NAc; (vi) hippocampal DOPAL and DOPAC levels were increased in females but decreased in males.

The higher METH intake observed in male rats during the self-administration experiment is consistent with data from previous animal studies [25,28,60,61]. For example, Ruda-Kucerova et al. (2015) reported that male Sprague–Dawley rats self-administered more METH during the last 5 days during a total of 14 SA training days [25]. Job et al. (2020) also showed that male Long–Evans rats acquired METH quicker than females during 20 days of SA training [28]. In addition, adolescent male Wistar rats were reported to take more METH than females [61]. Moreover, male baboons puffed significantly more METH aerosols than females [60]. Those observations are consistent with clinical reports that men use more METH than women [9,16,17]. This pattern of METH intake in men also caused more emergency department admissions [62,63,64] and higher numbers of METH poisoning cases and overdose deaths [7,8,10,12,65].

Because DA systems play an important role in the clinical course of several substance use disorders [66,67,68,69], efforts have been made to understand the consequences of drug use on those systems in humans. For example, it has been reported that dysregulation of DA systems in human METH users is associated with a higher prevalence of neuropsychiatric problems in those patients [70,71,72]. Although animal studies have tended to replicate evidence for dopaminergic dysfunctions after METH [72,73,74], investigators evaluating the impact of sex differences in METH-induced damage to dopaminergic systems have mainly used non-contingent administration of toxic METH doses [49,50,51,75]. We have therefore used the SA model to investigate these issues further. In the present METH self-administration study, our observation that female rats exhibited significant decreases in the levels of cortical DA and its metabolites (DOPAL and DOPAC) are comparable with results reported by Szczepanik et al. (2020), who found methylglyoxal-induced decreased DA levels in the PFC of female mice that exhibited memory impairment and depressive-like behaviors [76]. Together, these results suggest that METH-induced decreased DA levels might be, in part, responsible for depression [20,22] and memory impairments [77] reported in female METH users. Nevertheless, these notions need to be further examined in patient populations.

We also found differential METH SA-induced changes in DA metabolites in female rats. These differences might also be responsible, in part, for some of the memory deficits observed in female METH users [71,77]. Experiments are needed to investigate these issues further. Accumulation of the oxidative DA metabolites, DOPAL and DOPAC, in the hippocampus of female rats suggests that there might be increased DA turnover and associated increased production of reactive oxygen species (ROS) [78,79] with secondary damage to hippocampal structures [15,80] and associated memory impairments as reported previously in female METH users [77]. The observations of decreased hippocampal levels of DA and of its metabolite, DOPAL, indicate that the consequences of METH in that structure are sexually divergent, although the potential bases for these differences are not clear at present. Nevertheless, the data in males are consistent with those of the report of Almalki et al. (2018), who had reported that injections of METH (10 mg/kg i.p. 2 h × 4) caused decreased hippocampal DA levels in rats [81].

Unlike the observations made in the mesocorticolimbic DA pathway, METH appeared to have had minimal impact on the nigrostriatal DA system of these rats, consistent with the report of Schwendt et al. (2009), who also failed to observe any changes in striatal levels of DA and DOPAC in male rats withdrawn from METH SA for 15 days [82]. Similarly, other studies had not found any changes in DOPAC levels after 14 or a month of withdrawal [47,48].

## 4. Materials and Methods

### 4.1. Animals and SA Procedures

METH dose and SA training procedures were used as previously described [27]. Briefly, female (n = 24, 350–500 g) and male (n = 24, 450–600 g) Long–Evans rats were anesthetized using ketamine and xylazine (50 and 5 mg/kg, i.p., respectively). A Silastic catheter was then implanted into the right jugular vein as described previously [27]. Both female and male rats used in the study were drug-naïve and were not food-trained before the start of the SA experiment. After recovery from surgery, rats were randomly assigned to either METH (n = 18) or saline (n = 6) groups and were trained to self-administer METH (0.1 mg/kg/infusion, i.v.) for two 3 h sessions/day (separated by a 30 min off interval) for 20 days under FR-1 schedule with 20 sec timeouts after each infusion in Med Associates SA chambers. We trained rats to self-administer METH for 5 days per week with weekends off. On weekends, all animals remained in SA chambers but were disconnected from i.v. SA lines. Control rats self-administered saline under similar conditions. There was no limit on the number of infusions per 3 hr session, but both sessions were separated by 30 min breaks (during which the house light was turned off and the active lever was retracted) to prevent overdose.

After 20 days of SA training, rats were individually housed in the animal care facility for 30 days with no access to METH. Cue-induced drug seeking was also quantified on withdrawal days 3 (WD3) and 30 (WD30). The cue-induced seeking consisted of a single 3 hr session. All rats tested on WD3 were also tested on WD30. During the test, a press on the METH-associated lever accompanied the presentation of the tone and light cues previously paired with METH infusions but did not result in a METH infusion. All the animal procedures were approved and conducted according to the Guide for the Care and Use of Laboratory Animals (ISBN 0-309-05377-3) by the National Institute of Drug Abuse Animal Care and Use Committee (NIDA-ACUC).

### 4.2. Tissue Collection

To evaluate potential sex differences in the levels of DA and its metabolites after drug SA and abstinence, rats were euthanized 24 hr after WD30 by rapid decapitation with a guillotine. Dorsal striatum (dSTR, A/P +2 to −2 mm bregma, M/L ± 2 to 5 mm, D/V −3 to −6 mm), nucleus accumbens (NAc, A/P + 2.7 to + 0.7 mm bregma, M/L + 0.6 to + 2.2 mm, D/V + 5.6 to + 7.6 mm), prefrontal cortex (PFC, A/P + 2.7 to + 1.7 mm bregma, M/L 0 to + 4 mm, D/V + 7 to + 9 mm), and hippocampus (HIP, A/P −5 to −7 mm bregma, mediolateral ±6 mm, D/V −2 to −8 mm) were dissected out using specific neuroanatomical coordinates based on a rat atlas and immediately snap-frozen on dry ice and stored at −80 °C.

### 4.3. Measurement of Dopamine and Metabolites

The levels of DA and its metabolites were measured using high-performance liquid chromatography (HPLC) as per previously published protocols [83,84,85,86]. Briefly, frozen tissue samples were homogenized using a Branson sonifier 150 in a solution of 20:80 of 0.035 M phosphoric acid:0.2 M acetic acid using a ratio of 50 mg tissue to 250 uL of 20:80. The supernatant was assayed by batch alumina extraction followed by HPLC with Waters 515 pump, Water 717 autosampler (Waters Corporation, Milford, MA, USA), and ESA Choulochem 3 electrochemical detector (ESA, Inc. Chelmsford MA) with a series of electrochemical detection, Cera column temperature controller 250 (Cera, Inc., Baldwin Park, CA, USA) set to 18 degrees using a Spheri-5 RP-18, 5 μm, 30 × 4.6 mm guard column (PerkinElmer, MA, USA, no. 07110013), and Bio-advantage C18, 5 µm, 120 Å, 4.6 × 250 mm analytic column (Thomson Instruments, Clear Brook, VA, USA, No. BA400-046250). The mobile phase consisted of 13.8 g monobasic sodium phosphate, 64 mg octane sulfonic acid, 50 mg EDTA, and 25–30 mL acetonitrile in 1 L of HPLC-grade water, adjusted pH to 3.15–3.25 using 85% phosphoric acid. Concentrations of DA and its metabolites in brain regions were expressed in units of picomoles per mg per weight.

### 4.4. Statistical Analyses

Effects of daily METH intake, withdrawal day (active lever presses), and levels of DA and its metabolites were analyzed using 2-way ANOVA followed by Fisher’s protected least significance difference post-hoc test using GraphPad Prism 9. For all analyses, the null hypothesis was rejected at *p* ≤ 0.05.

## 5. Conclusions

In conclusion, we found sex differences in the levels of DA and its metabolites in the PFC, NAc, and HIP (Table 1 and Table 2). Female rats that self-administered METH showed decreased DA levels in the PFC, suggesting that the levels of METH might have reached toxic levels in the PFC of female but not male rats. This suggestion is consistent with the report that female rats had higher brain levels of amphetamine after systemic injections [52]. In addition, male high METH takers exhibited decreased DA levels in the HIP. Together, these observations suggest potential important relationships between the effects of METH SA on mesocorticolimbic dopaminergic systems and behavioral responses during the process of withdrawal from METH. This reasoning might not apply to the nigrostriatal DA system since we observed minimal changes in striatal DA and its metabolites. Our results further support the notion that more efforts need to be spent to identify potential roles that sex might play in the clinical manifestations of METH use disorder. Given the major sex differences that we have identified in the PFC, NAc, and HIP of rats that had self-administered, it is not farfetched to suggest that similar consequences might occur in the brains of women and men who self-administer methamphetamine. This conclusion is consistent with the clinical presentations discussed in the introduction and discussion of this paper. Finally, our results suggest that sex-relevant behavioral and pharmacological approaches are essential when dealing with humans who meet the criteria for METH use disorder because the behavioral and biochemical consequences of METH exposure appear to be sexually divergent.

## Figures and Tables

**Figure 1 ijms-23-04353-f001:**
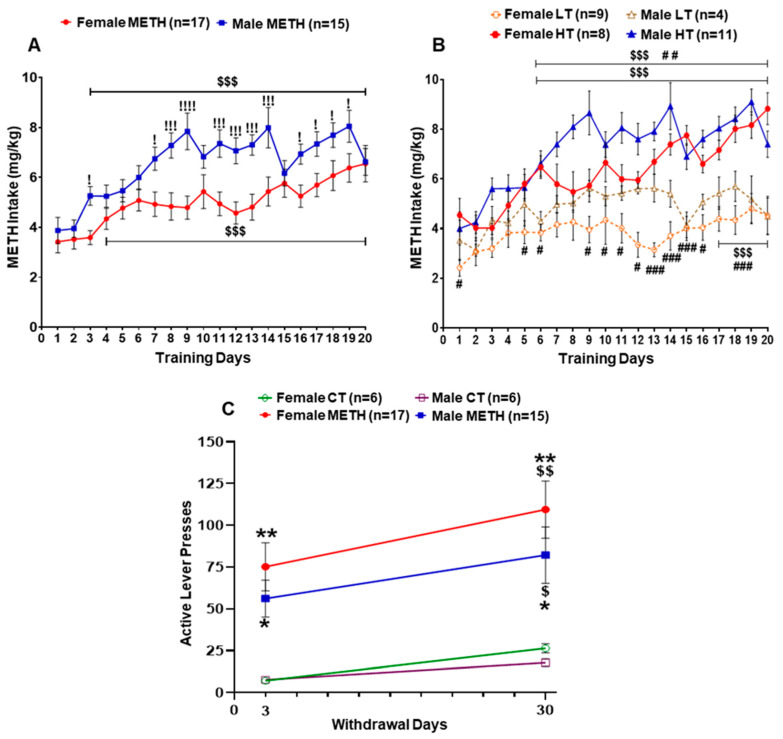
Behaviors of female and male rats after METH SA and during METH-seeking tests. (**A**) Patterns of daily METH intake behaviors by female and male rats. (**B**) METH intake by female and male low and high METH takers. (**C**) Drug-seeking behaviors by female and male rats at WD3 and WD30. Key to statistics: * *p* < 0.05, ** *p* < 0.01, female and male METH groups compared with respective saline groups; ! *p* < 0.05, !!! *p* < 0.001, comparison of male METH group with female METH group; $ *p* < 0.05, $$ *p* < 0.01, $$$ *p* < 0.001, comparison of daily METH intake of female and male rats compared to METH intake on the first day of SA or comparison of active lever pressing on WD30 compared to WD3. # *p* < 0.05, ## *p* < 0.01, ### *p* < 0.001, comparison between female or male high METH takers vs. low METH takers. All values represent means ± SEM of number of animals indicated in the figure. CT, control saline rats; LT, low METH takers; HT, high METH takers.

**Figure 2 ijms-23-04353-f002:**
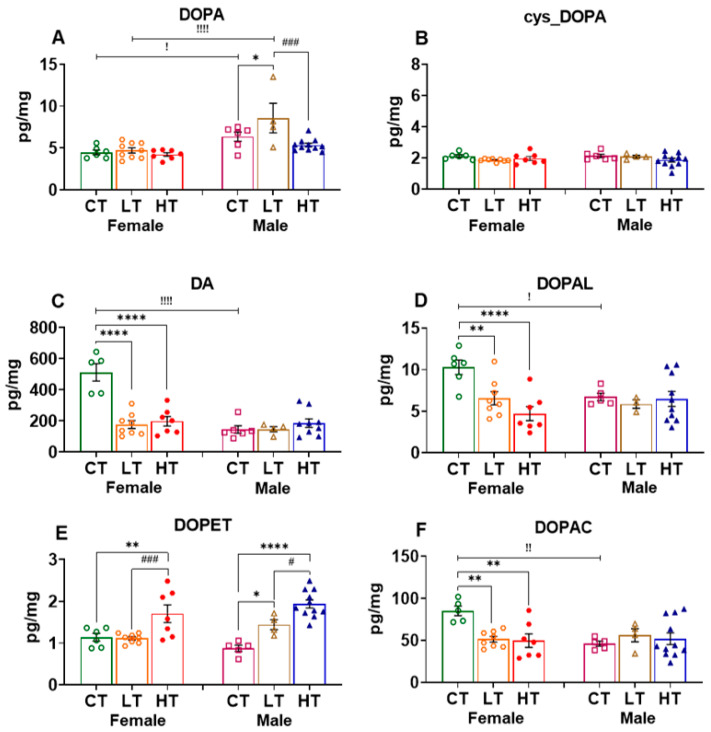
Levels of dopamine and its metabolites in the prefrontal cortex (PFC). (**A**) DOPA, (**B**) cys_DOPA, (**C**) DA, (**D**) DOPAL, (**E**) DOPET, and (**F**) DOPAC in female and male METH rats compared to controls after 30 days of withdrawal from METH SA. Key to statistics: * *p* < 0.05, ** *p* < 0.01, **** *p* < 0.0001, comparison between controls vs. low and high METH-taker groups; # *p* < 0.05, ### *p* < 0.001, comparison between low and high METH-taker groups; ! *p* < 0.05, !! *p* < 0.01, !!!! *p* < 0.0001, comparison between female and male rats. All values represent means ± SEM of number of animals indicated in the figure. CT, control saline rats; LT, low METH takers; HT, high METH takers.

**Figure 3 ijms-23-04353-f003:**
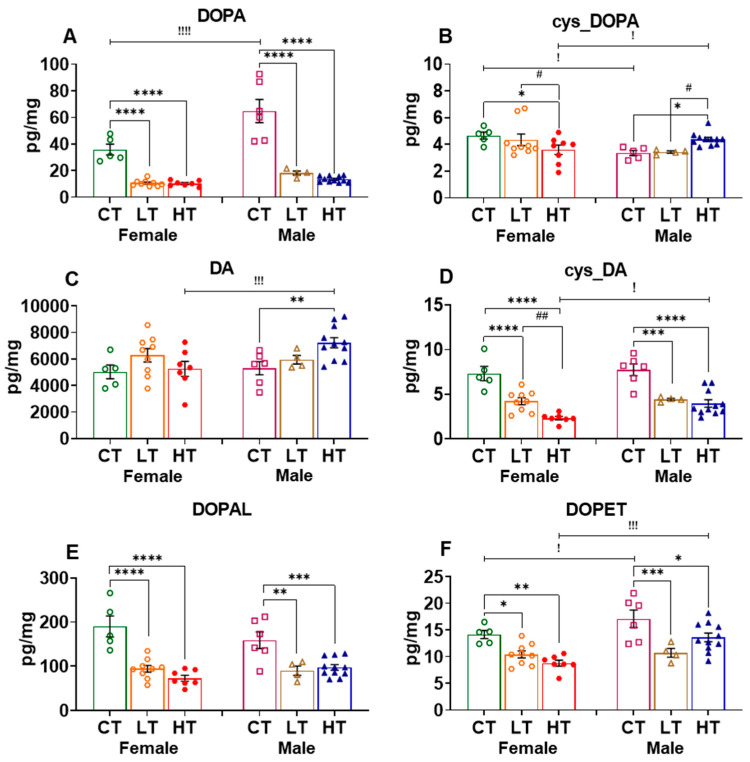
Content of dopamine and its metabolites in the nucleus accumbens (NAc) after 30 days of withdrawal from METH SA. (**A**) DOPA, (**B**) cys_DOPA, (**C**) DA, (**D**) cys_DA, (**E**) DOPAL, (**F**) DOPET, and (**G**) DOPAC in female and male rats. * *p* < 0.05, ** *p* < 0.01, *** *p* < 0.001, **** *p* < 0.0001. # *p* < 0.05, ## *p* < 0.01. ! *p* < 0.05, !! *p* < 0.01, !!! *p* < 0.001, !!!! *p* < 0.0001.

**Figure 4 ijms-23-04353-f004:**
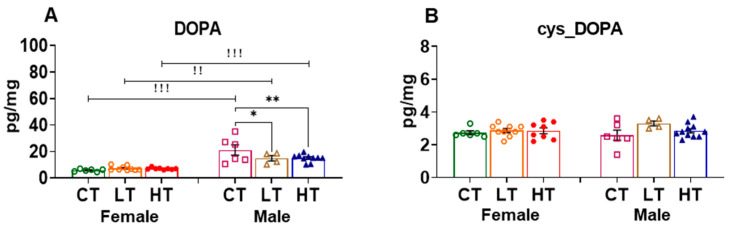
Effect of METH SA and withdrawal in the dorsal striatum (dSTR) dopamine and its metabolites level. (**A**) DOPA, (**B**) cys_DOPA, (**C**) DA, (**D**) cys_DA, (**E**) DOPAL, (**F**) DOPET, and (**G**) DOPAC in female and male rats. * *p* < 0.05, ** *p* < 0.01. # *p* < 0.05. ! *p* < 0.05, !! *p* < 0.01, !!! *p* < 0.001.

**Figure 5 ijms-23-04353-f005:**
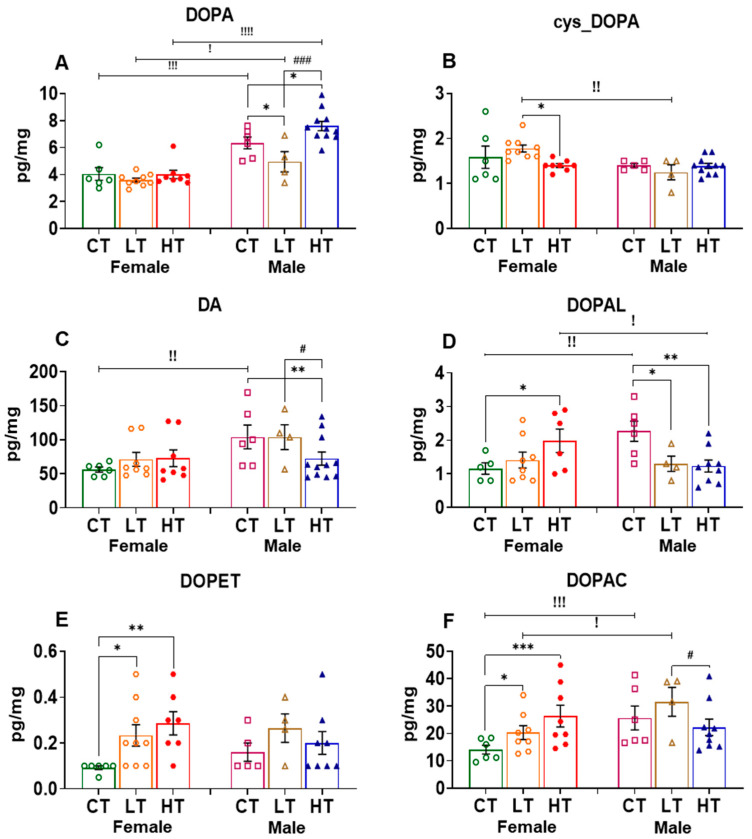
Level of dopamine and its metabolites in the hippocampus (HIP). (**A**) DOPA, (**B**) cys_DOPA, (**C**) DA, (**D**) DOPAL, (**E**) DOPET, and (**F**) DOPAC in female and male METH rats after 30 days of withdrawal from METH SA. * *p* < 0.05, ** *p* < 0.01, *** *p* < 0.001. # *p* < 0.05, ### *p* < 0.001. ! *p* < 0.05, !! *p* < 0.01, !!! *p* < 0.001.

**Table 1 ijms-23-04353-t001:** Sex differences in dopamine and its metabolites in the brain.

Effect of Sex	Female vs. Male											
	PFC			NAc			dSTR			HIP		
	CT	LT	HT	CT	LT	HT	CT	LT	HT	CT	LT	HT
**DOPA**	*↓*	**↓↓↓**	NS	**↓↓↓**		NS	**↓↓↓**	**↓↓**	**↓↓↓**	**↓↓↓**	**↓**	**↓↓↓**
**cys_DOPA**	NS	NS	NS	**↑**	NS	**↓**	NS	NS	NS	NS	**↑↑**	NS
**DA**	**↑↑↑**	NS	NS	NS	NS	**↓↓↓**	NS	NS	NS	**↓↓**	NS	NS
**cys_DA**	ND	ND	ND	NS	NS	↓	NS	↓↓	↓	ND	ND	ND
**DOPAL**	**↑**	NS	NS	NS	NS	NS	NS	NS	NS	**↓↓**	NS	**↑**
**DOPET**	NS	NS	NS	**↑**	NS	**↓↓↓**	NS	NS	NS	NS	NS	NS
**DOPAC**	**↑↑**	NS	NS	**↓↓**	NS	**↓↓↓**	NS	NS	NS	**↓↓↓**	**↓**	NS

Abbreviation: PFC: prefrontal cortex, NAc; nucleus accumbens, dSTR; dorsal striatum, HIP; hippocampus, CT; control rats, LT; low METH takers, HT; high METH takers; DOPA; 3,4-dihydroxyphenylalanine, cys_DOPA; 5-S-cysteinyl-DOPA, DA; dopamine, cys_DA; 5-S-cysteinyl-DA, DOPAL; 3,4-Dihydroxyphenylacetaldehyde, DOPET; 3,4-dihydroxyphenylethanol, DOPAC; 3,4-dihydroxyphenylacetic acid, ↑ (*p* < 0.05), ↑↑ (*p* < 0.05), ↑↑↑ (*p* < 0.001); higher level: ↓ (*p* < 0.05), ↓↓ (*p* < 0.01), ↓↓↓ (*p* < 0.001); lower level. ND; not detected; NS: non-significant.

**Table 2 ijms-23-04353-t002:** Effects of METH SA on dopamine and its metabolites.

Effect of METH SA	PFC						NAc						dSTR						HIP					
	Female			Male			Female			Male			Female			Male			Female			Male		
	CT vs. LT	CT vs. HT	HT vs. LT	CT vs. LT	CT vs. HT	HT vs. LT	CT vs. LT	CT vs. HT	HT vs. LT	CT vs. LT	CT vs. HT	HT vs. LT	CT vs. LT	CT vs. HT	HT vs. LT	CT vs. LT	CT vs. HT	HT vs. LT	CT vs. LT	CT vs. HT	HT vs. LT	CT vs. LT	CT vs. HT	HT vs. LT
**DOPA**	NS	NS	NS	↑	NS	↑↑↑	↓↓↓	↓↓↓	NS	↓↓↓	↓↓↓	NS	NS	NS	NS	↓	↓↓	NS	NS	NS	NS	↓	↑	↑↑↑
**cys_DOPA**	NS	NS	NS	NS	NS	NS	NS	↓	↓	NS	↑	↑	NS	NS	NS	NS	NS	NS	NS	NS	↓	NS	NS	NS
**DA**	↓↓↓	↓↓↓	NS	NS	NS	NS	NS	NS	NS	NS	↑	NS	NS	NS	NS	NS	NS	NS	NS	NS	NS	NS	↓↓	↓
**cys_DA**	ND	ND	ND	ND	ND	ND	↓↓↓	↓↓↓	↓↓	↓↓↓	↓↓↓	NS	NS	NS	NS	NS	NS	↓	ND	ND	ND	ND	ND	ND
**DOPAL**	↓↓	↓↓↓	NS	NS	NS	NS	↓↓↓	↓↓↓	NS	↓↓↓	↓↓↓	NS	NS	NS	NS	NS	NS	NS	NS	↑	NS	↓	↓↓	NS
**DOPET**	NS	↑↑↑	↑↑	↑	↑↑↑	↑	↓	↓↓	NS	↓↓↓	↓	NS	NS	NS	NS	NS	NS	NS	↑	↑↑	NS	NS	NS	NS
**DOPAC**	↓↓	↓↓	NS	NS	NS	NS		↓	↓↓	↓↓	↓		NS	NS	NS	NS	NS	NS	↑	↑↑↑	NS	NS	NS	↓

Abbreviation: PFC: prefrontal cortex, NAc; nucleus accumbens, dSTR; dorsal striatum, HIP; hippocampus, CT; control rats, LT; low METH takers, HT; high METH takers; DOPA; 3,4-dihydroxyphenylalanine, cys_DOPA; 5-S-cysteinyl-DOPA, DA; Dopamine, cys_DA; 5-S-cysteinyl-DA, DOPAL; 3,4-Dihydroxyphenylacetaldehyde, DOPET; 3,4-dihydroxyphenylethanol, DOPAC; 3,4-dihydroxyphenylacetic acid, ↑ (*p* < 0.05), ↑↑ (*p* < 0.05), ↑↑↑ (*p* < 0.001); higher level: ↓ (*p* < 0.05), ↓↓ (*p* < 0.01), ↓↓↓ (*p* < 0.001); lower level. ND: not detected; NS: non-significant.

## Data Availability

Not applicable.

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
