# Peer review of "Sex-Specific Alterations in Dopamine Metabolism in the Brain after Methamphetamine Self-Administration"

_ijms, 2022, doi:10.3390/ijms23084353_

Round 1

Reviewer 1 Report

I've found the present study really well conducted and impressive. Authors found sex differences in the levels of DA and its metabolites in the PFC, NAc, and HIP. These observations suggest potential important relationships between the effects of METH SA on mesocorticolimbic dopaminergic systems and behavioral responses after drug withdrawal. 

The study is complete and well-written, I only suggest to enrich the Conclusions section by adding potential implications of the fact that sex might play in the clinical manifestations of METH use disorder and its potential treatment.

Author Response

Responses to Reviewer comments

 We thank the editors and reviewers for their constructive comments that help to greatly improve our manuscript.

Reviewer # 1:

Q1. 1.  The study is complete and well-written. I only suggest enriching the Conclusions section by adding potential implications of the fact that sex might play in the clinical manifestations of METH use disorder and its potential treatment.

Response. We thank the reviewer for the positive comments. As per the suggestion, we have included more information related to the treatment of the clinical manifestations of METH use disorders (see line 449-466). 

Reviewer 2 Report

The goal of this experiment was to examine lingering sex differences in dopamine and its metabolites in several drug abuse-related regions of the brain following a period of methamphetamine self-administration and 30 day abstinence.  The experiment is well-designed and executed, and the manuscript is a pleasure to read.  My main criticism of the paper is that the discussion goes far afield from the data, bordering on idle speculation.

Main Concerns:

1) A limitation of this and other sex difference papers of its type is the contribution of differential brain levels of the drug following systemic administration.  Decades ago, Becker et al. (1982) demonstrated higher brain levels of amphetamine in female rats than male rats following comparable systemic dosing.  These measurements are certainly difficult to obtain, but they can factor into apparent sex differences that amount to dosing differences rather than intrinsic differences in neurotransmission/plasticity.

2) The discussion is placed in the context of methamphetamine use disorder, though there was no evaluation of the development of compulsive drug seeking.  Perhaps the discussion could be toned down in this regard.

3) The discussion of behavioral pathologies based on the regions affected by the methamphetamine is rather weak.  The authors refer to several behavioral pathologies following methamphetamine use (e.g., anxiety, psychosis, poor decision making), but there were no measurements made in these rats to suggest that the methamphetamine self-administration altered behavior modeling these disorders.  Thus it is inappropriate to attribute the changes in dopamine and its metabolites to mechanisms underlying these pathologies.  This is further problematic as there is not clear evidence that alterations in dopamine neurotransmission would underlie these behavioral pathologies vs. other transmitter systems (e.g., glutamate).

Minor Points:

1) In line 9 in the abstract, I am not sure dimorphism is appropriate.  Perhaps replace with 'sex differences'?

2) In the section on Tissue Collection, more detail of the dissection (e.g., thickness?) and landmarks used to isolate the regions is needed.

3) The dopaminergic endpoints are extensive.  A summary table of the results including sex, brain region, and neurochemical measurement would be helpful for the reader.

Author Response

Responses to Reviewer comments

 We thank the editors and reviewer for their constructive comments that help to greatly improve our manuscript.

Reviewer # 2:

Q1. A limitation of this and other sex difference papers of its type is the contribution of differential brain levels of the drug following systemic administration.  Decades ago, Becker et al. (1982) demonstrated higher brain levels of amphetamine in female rats than male rats following comparable systemic dosing. These measurements are certainly difficult to obtain, but they can factor into apparent sex differences that amount to dosing differences rather than intrinsic differences in neurotransmission/plasticity.

Response. This point is well taken. Nevertheless, this comment does not contradict the existence of sex differences in the presentation or consequences of the use of amphetamine-type drugs. We have included a statement to that effect and have cited the mentioned paper in our discussion (see line 141-146 and 242-244). 

Q2. The discussion is placed in the context of methamphetamine use disorder, though there was no evaluation of the development of compulsive drug seeking. Perhaps the discussion could be toned down in this regard.

Response. As suggested, we have toned down the discussion (see lines 28-30, 343-346 and 351-357).

Q3.  The discussion of behavioral pathologies based on the regions affected by the methamphetamine is rather weak. The authors refer to several behavioral pathologies following methamphetamine use (e.g., anxiety, psychosis, poor decision making), but there were no measurements made in these rats to suggest that the methamphetamine self-administration altered behavior modeling these disorders. Thus it is inappropriate to attribute the changes in dopamine and its metabolites to mechanisms underlying these pathologies.  This is further problematic as there is not clear evidence that alterations in dopamine neurotransmission would underlie these behavioral pathologies vs. other transmitter systems (e.g., glutamate).

Response. We addressed the point raised by reviewer and made necessary changes in the manuscript (see lines 343-346 and 351-357).

Q4.  In line 9 in the abstract, I am not sure dimorphism is appropriate.  Perhaps replace with 'sex differences'?.

Response. We fixed that (see line 9).

Q5.  In the section on Tissue Collection, more detail of the dissection (e.g., thickness?) and landmarks used to isolate the regions is needed.

Response. Neuroanatomical coordinates are now provided in the method section (see line 414-418).

Q6.  The dopaminergic endpoints are extensive.  A summary table of the results including sex, brain region, and neurochemical measurement would be helpful for the reader.

Response. This is a valuable suggestion. We have now included two tables (tables 1 and 2) that summarize all the results (see line 367-381).